# Transformer Module Networks for Systematic Generalization in Visual Question Answering

## Abstract

Transformers achieve great performance on Visual Question Answering (VQA). However, their systematic generalization capabilities, i.e., handling novel combinations of known concepts, is unclear. We reveal that Neural Module Networks (NMNs), i.e., question-specific compositions of modules that tackle a sub-task, achieve better or similar systematic generalization performance than the conventional Transformers, even though NMNs' modules are CNN-based. In order to address this shortcoming of Transformers with respect to NMNs, in this paper we investigate whether and how modularity can bring benefits to Transformers. Namely, we introduce Transformer Module Network (TMN), a novel NMN based on compositions of Transformer modules. TMNs achieve state-of-the-art systematic generalization performance in three VQA datasets, improving more than 30% over standard Transformers for novel compositions of sub-tasks. We show that not only the module composition but also the module specialization for each sub-task are the key of such performance gain.

## 1 Introduction

Visual Question Answering (VQA) (Antol et al., 2015) is a fundamental testbed to assess the capability of learning machines to perform complex visual reasoning. The compositional structure inherent to visual reasoning is at the core of VQA: Visual reasoning is a composition of visual sub-tasks, and also, visual scenes are compositions of objects, which are composed of attributes such as textures, shapes and colors. This compositional structure yields a distribution of image-question pairs of combinatorial size, which cannot be fully reflected in an unbiased way by training distributions.

Systematic generalization is the ability to generalize to novel compositions of known concepts beyond the training distribution (Lake & Baroni, 2018; Bahdanau et al., 2019; Ruis et al., 2020). A learning machine capable of systematic generalization is still a distant goal, which contrasts with the exquisite ability of current learning machines to generalize in-distribution. In fact, the most successful learning machines, i.e., Transformer-based models, have been tremendously effective for VQA when evaluated in-distribution (Tan & Bansal, 2019; Chen et al., 2020; Zhang et al., 2021). Yet, recent studies stressed the need to evaluate systematic generalization instead of in-distribution generalization (Gontier et al., 2020; Tsarkov et al., 2021; Bergen et al., 2021), as the systematic generalization capabilities of Transformers for VQA are largely unknown.

A recent strand of research for systematic generalization in VQA investigates Neural Module Networks (NMNs) (Bahdanau et al., 2019; 2020; D'Amario et al., 2021). NMNs decompose a question in VQA into sub-tasks, and each sub-task is tackled with a shallow neural network called *module*. Thus, NMNs use a question-specific composition of modules to answer novel questions. NMNs alleviate the gap between in-distribution generalization and systematic generalization due to its inherent compositional structure. In our experiments, we found that CNN-based NMNs outperform Transformers on systematic generalization to novel compositions of sub-tasks. This begs the question of whether we can combine the strengths of Transformers and NMNs in order to improve the systematic generalization capabilities of learning machines.

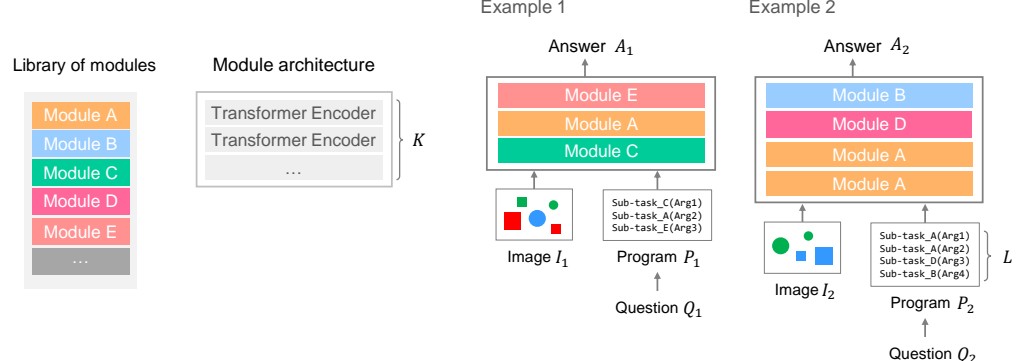

(a) Library and module architecture    (b) Examples of question-specific networks

Figure 1: *Overview of Transformer Module Network (TMN)*. (a) TMN has a library of neural modules that tackle a sub-task, and each module is implemented with a small stack of Transformer encoders. (b) The modules are composed into a question-specific network. This network takes as input an image and a program that corresponds to a question. The total number of Transformer layers is the number of Transformer layers of each module $(K) \times$ the program length $(L)$.

In this paper, we introduce *Transformer Module Network (TMN)*, a novel NMN for VQA based on compositions of Transformer modules. In this way, we take the best of both worlds: the capabilities of Transformers given by attention mechanisms, and the flexibility of NMNs to adjust to questions based on novel compositions of modules. TMN allows us to investigate whether and how modularity brings benefits to Transformers in VQA. An overview of TMNs is depicted in Fig. 1.

To foreshadow the results, we find that TMNs achieve state-of-the-art systematic generalization accuracy in the following three VQA datasets: CLEVR-CoGenT (Johnson et al., 2017), CLO-SURE (Bahdanau et al., 2020), and a novel test set based on GQA (Hudson & Manning, 2019) that we introduce for evaluating systematic generalization performance with natural images, which we call GQA-SGL (Systematic Generalization to Linguistic combinations). Remarkably, TMNs improve systematic generalization accuracy over standard Transformers more than $30\%$ in the CLO-SURE dataset, i.e., systematic generalization to novel combinations of known linguistic constructs (equivalently, sub-tasks). Our results also show that both module composition and module specialization to a sub-task are key to TMN's performance gain.

## 2    RELATED WORK

We review previous works on systematic generalization in VQA. We first revisit the available benchmarks and then introduce existing approaches.

**Benchmarking VQA.** Even though systematic generalization capabilities are the crux of VQA, attempts to benchmark these capabilities are only recent. The first VQA datasets evaluated in-distribution generalization, and later ones evaluated generalization under distribution shifts that do not require systematicity. In the following, we review progress made towards benchmarking systematic generalization in VQA:

*–In-distribution generalization:*   There is a plethora of datasets to evaluate in-distribution generalization, e.g., VQA-v2 (Goyal et al., 2019) and GQA (Hudson & Manning, 2019). It has been reported that these datasets are biased and models achieve high accuracy by relying on spurious correlations instead of performing visual reasoning (Agrawal et al., 2018; Kervadec et al., 2021).

*–Out-of-distribution generalization:*   VQA-CP (Agrawal et al., 2018) and GQA-OOD (Kervadec et al., 2021) were proposed to evaluate generalization under shifted distribution of question-answer pairs. While this requires a stronger form of generalization than in-distribution, it does not require tackling the combinatorial nature of visual reasoning, and models can leverage biases in the images and questions.

*–Systematic generalization:* CLEVR-CoGenT (Johnson et al., 2017) and CLOSURE (Bahdanau et al., 2020) are datasets that require systematic generalization as models need to tackle novel combinations of visual attributes and sub-tasks. Since these datasets include only synthetic images, we introduce GQA-SGL, a novel test set based on GQA to evaluate systematic generalization with natural images.

**Approaches for Systematic Generalization.** We now revisit Transformer-based models and NMNs for systematic generalization in VQA as they are the basis of TMNs:

*–Transformer-Based Models:* Currently, most approaches to VQA are based on Transformers (Vaswani et al., 2017), e.g., Tan & Bansal (2019); Chen et al. (2020); Zhang et al. (2021), and pre-training is at the core of these approaches. These works focus on in-distribution generalization, and it was not until recently that Transformers for systematic generalization have been investigated. To the best of our knowledge, in VQA the only related work is MDETR (Kamath et al., 2021), which uses a novel training approach that captures the long tail of visual concepts and achieves state-of-the-art performance on many vision-and-language datasets, including CLEVR-CoGenT. As we show in the sequel, our approach shows better systematic generalization capabilities in CLEVR-CoGenT and CLOSURE without requiring pre-trained Transformer encoders.

*–Neural Module Networks (NMNs):* They represent a question in the form of a program in which each sub-task is implemented with a neural module. Thus, modules are composed into a network specific for the question (Andreas et al., 2016). Some of the most successful modular approaches include NS-VQA (Yi et al., 2018), which uses a symbolic execution engine instead of modules based on neural networks, and the Meta-Module Network (MMN) (Chen et al., 2021), which introduces a module that can adjust to novel sub-tasks. While these approaches are effective for in-distribution generalization, they fall short in terms of systematic generalization. Vector-NMN (Bahdanau et al., 2020) is the state-of-the-art NMN on systematic generalization and outperforms all previous modular approaches. We present evidence that the systematic generalization capabilities of modular approaches can be improved by using the attention mechanisms of Transformer architectures.

## 3 TRANSFORMER MODULE NETWORKS (TMNS)

In this section we introduce TMNs, i.e., our novel Transformer-based model for VQA that composes neural modules into a question-specific Transformer network. Unlike previous NMNs, TMNs regard the tokens that represent the input image as a workspace shared among the modules, which transform the token representations. This architectural change is not straightforward and thus we need to architect NMNs taking into account the philosophy of tokens to leverage the attention mechanism.

**Question-specific Composition of Modules.** Figure 1 provides an overview. TMNs use a library of modules, and each module tackles a different sub-task we defined (e.g., `FILTER`, `COUNT`). TMNs represent the question in VQA with a program, which is a sequence of sub-tasks each of which is implementable with a module from the library. For instance, a question "How many red squares are there?" can be converted to a program $\{$ `FILTER(square)`, `FILTER(red)`, `COUNT` $\}$, and then the modules corresponding to the sub-tasks are composed to form a network to answer the question.

In TMN, each module is a stack of Transformer encoders. Unlike the standard Transformers, which are a stack of a fixed number of Transformer encoders (typically 12) regardless of the input question, TMN composes the Transformer modules into a question-specific network, and thus, the total number of encoder layers in TMN varies according to the question. Let $K$ be the number of Transformer encoder layers in each module, and let $L$ be the number of sub-tasks in the program. Then, TMN has a total of $K \times L$ layers to implement a program of length $L$.

We follow the same approach in previous works that directly use the program rather than the raw question (Bahdanau et al., 2020; 2019; D'Amario et al., 2021). This allows to analyse systematic generalization in isolation by focusing only on visual reasoning aspects, and omits the performance of the language parser.

**Transformer-based Architecture.** The aforementioned question-specific composition of modules is preceded by a feature extractor and followed by an output classifier, as shown in Fig. 2(a).

We use a feature extractor to obtain the visual features of the input image. We follow the common procedure in Transformers: The visual features are transformed into the visual feature embeddings

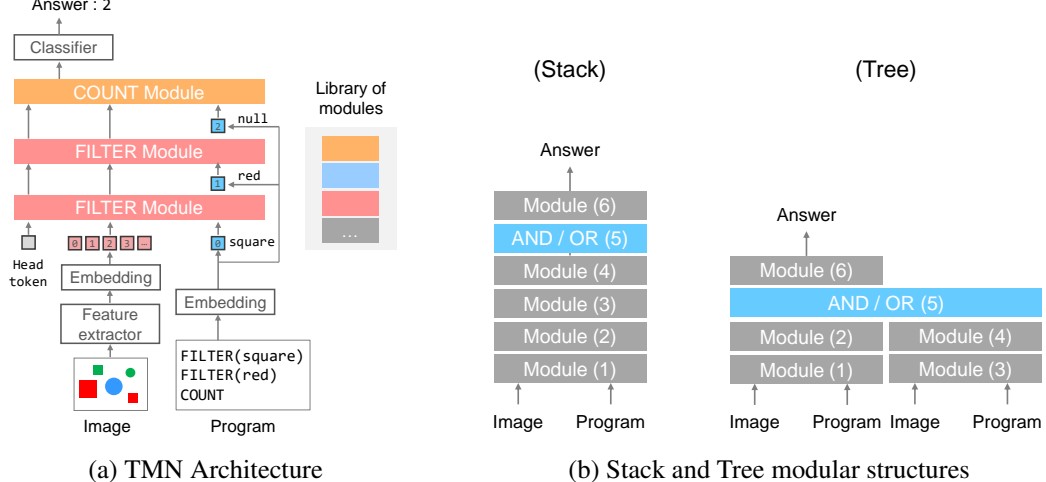

(a) TMN Architecture

(b) Stack and Tree modular structures

Figure 2: *TMN Architecture and Modular Structures.* (a) Given an image and a program related to a question, TMNs construct a question-specific Transformer network as a composition of modules that tackle a sub-task. Each module takes as input the representations of the visual features, the head token and the module's argument from the program. A classifier outputs an answer from the final representation of the head token. (b) We investigate two modular structures. Stack structure simply stacks all the modules. Tree structure runs two threads and then merges their outputs with a module (`AND`/`OR` in this example). The numbers in ( ) denotes the order of each sub-task in the program.

and then summed with the position embedding, which yields the input visual representations of the first module. The initial head token representation is the average of all the visual representations.

Then, the first Transformer module in the question-specific network takes as input the sequence of tokens consisting of the representations of the visual features, the head token, and the module's argument coming from the program. The Transformer module outputs the transformed representations of the head token and visual features. This output is fed into the next module, which also receives its corresponding argument coming from the program. Unlike Vector-NMN, whose modules always receive the initial visual representations, in TMN only the first module receives the initial visual representations. Also, due to the versatility of Transformer architectures, in TMN we can easily change the number of Transformer encoders in each module according to the complexity of the sub-task.

Finally, a classifier outputs an answer from the final representation of the head token.

**Stack and Tree Modular Structures.** We further analyze the effect of modular structures. Some modules with two inputs may be useful for constructing a network corresponding to a complex program. For instance, a logical sub-task (`AND` or `OR`) or sub-tasks that compare two objects. For programs containing such sub-tasks, we investigate two modular structures exemplified in Fig. 2(b). Stack and tree structures are widely studied in previous works (Andreas et al., 2016; Hu et al., 2017). The stack structure simply stacks all modules and executes the modules in sequence. The tree structure runs two threads in parallel and then merges their outputs with a module (`AND`/`OR` in this example), which takes two sets of representations of visual features and a head token as an input and outputs the transformed representations of the first set in our case. We compare these two structures in the experiments.

## 4 EXPERIMENTAL SETUP

We use three datasets for VQA: two exising datasets and a novel test set we built. We compare our TMN with two Transformer-based baselines and several state-of-the-art models on those datasets.

**Datasets.** To evaluate the models for systematic generalization to novel combinations of known visual attributes or linguistic constructs, we use the following three datasets.

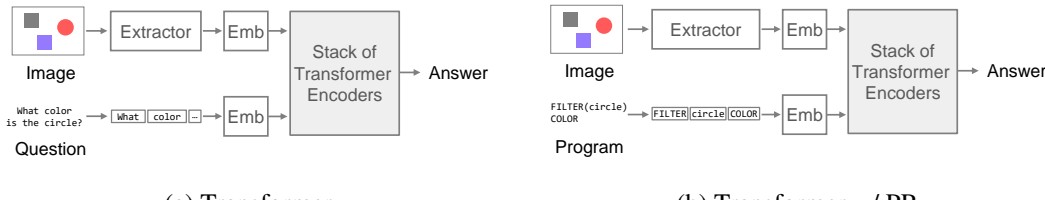

(a) Transformer            (b) Transformer w/ PR

Figure 3: *Transformer-based Baseline Methods.* (a) Standard Transformer (Transformer) takes an image and a question as inputs. (b) Standard Transformer with programs (Transformer w/ PR) takes a program as an input instead of the question. `Emb` denotes visual/word embeddings. These approaches use the same visual inputs and Transformer encoder structure.

–*CLEVR-CoGenT:*    CLEVR is a diagnostic dataset for VQA models (Johnson et al., 2017) that consists of synthetic 3D scenes with multiple objects and automatically generated questions, associated with a ground-truth program formed by multiple sub-tasks. This dataset comes with additional splits to test systematic generalization, namely, the Compositional Generalization Test (CLEVR-CoGenT). CLEVR-CoGenT is divided in two conditions where objects appear with limited color and shape combinations, that are inverted between training and testing. Using this dataset, we can test systematic generalization to novel combinations of visual attributes.

–*CLOSURE:*    This is a test set for models trained with CLEVR, and provides 7 benchmarks (Bahdanau et al., 2020). The benchmarks test the systematic generalization to novel combinations of known linguistic constructs, which can be considered as sub-tasks, e.g., "is the same size as" can be a sub-task `SAME_SIZE`. CLOSURE uses the same synthetic images in CLEVR but contains questions which require the models to recombine the known sub-tasks in a novel way.

–*GQA-SGL:*    GQA is a VQA dataset that consists of complex natural images and questions, associated with ground-truth programs (Hudson & Manning, 2019). GQA-SGL (Systematic Generalization to Linguistic combinations) is a novel test set we built based on GQA to test systematic generalization to novel combinations of known linguistic constructs with natural images. We generate new questions with ground-truth programs by combining the existing programs as shown in Fig. 4 in the Appendix A. To build the test set, we use the test-dev set from the balanced split. For the train set, in all experiments we always use the balanced split that controls bias. See Appendix A for details.

**Methods and Settings.** We compare TMNs with two Transformer-based baseline methods shown in Fig. 3. In (a), standard Transformer (Transformer) takes an image and a question as the inputs. In (b), standard Transformer with programs (Transformer w/ PR) takes a program as an input instead of the question and treat it as a sequence of tokens. TMNs also take the program as an input and executes each module with corresponding arguments as shown in Fig. 2. Thus, comparing Transformer w/ PR with TMNs allow us to assess the effectiveness of our Transformer modules and question-specific compositions of the modules. We treat VQA as a classification task and train the models with cross-entropy loss across the TMNs and our baselines.

–*Architecture:*    We use vanilla Transformer encoder for both standard Transformers and our TMNs for fair comparison. Standard Transformers consist of 12 Transformer layers. In TMNs, the number of Transformer layers in each module is 1 for CLEVR-CoGenT and CLOSURE, and 2 for GQA-SGL because of the difference in the complexity of their sub-tasks. Transformer encoders in all models have a hidden state size of 768 and 12 attention heads.

–*Sub-tasks and programs:*    For CLEVR-CoGenT and CLOSURE, we use the sub-tasks and programs defined in the dataset. For GQA-SGL, we follow the definitions proposed in Chen et al. (2021).

–*Visual features:*    For CLEVR-CoGenT and CLOSURE, we use grid features extracted by a ResNet-101 pre-trained with ImageNet to obtain the visual feature of $H \times W \times 2048$ dimensions, where $H$ and $W$ are the height and the width of the grid feature map. We add a positional encoding to the grid features and flatten them with a linear projection. We treat the flattened features as a sequence of tokens and thus the number of the tokens is $H \times W$ (150 on these datasets). The

input images in GQA contain various objects and are much more complex than those in CLOSURE. Therefore, for GQA-SGL, we use the popular regional features (Anderson et al., 2018) extracted by a Faster R-CNN (Ren et al., 2015) (with a ResNet-101 backbone) pre-trained on Visual Genome dataset (Krishna et al., 2016). We consistently keep 36 regions for each image. Each region has 2048-dimensional visual feature and 5-dimensional spatial location feature.

*–Embeddings:* The input representation of the program is a sum of word embeddings, segment embeddings and position embeddings. They represent a sub-task or an argument (e.g., `FILTER` or `circle`), a thread index, and word position, respectively. For the standard Transformer, we adopt BERT tokenizer and the standard sentence embeddings described in (Li et al., 2020).

*–Hyperparameters:* We use the Adam optimizer for all cases. Learning rates for standard Transformers and TMNs are 2e-5 and 1e-5 for CLEVR-CoGenT, 2e-5 for CLEVR, and 1e-5 and 4e-5 for GQA. We search best learning rates for each model. We train all the models with batch size of 128 on 4 Tesla V100 GPUs for 20 epochs except standard Transformer on CLEVR, where we use 30 epochs to achieve convergence. Training of standard Transformers and TMNs finished in 3 and 4 days, respectively. We use this experimental setup and hyperparameter tuning for all experiments.

*–Other methods:* We also experiment with other state-of-the-art methods, including Vector-NMN (Bahdanau et al., 2020), MDETR (Kamath et al., 2021) and MMN (Chen et al., 2021) (LXMERT (Tan & Bansal, 2019) in Appendix G). We use the hyperparamters described in the original papers. For a fair comparison, we train Vector-NMN and MMN using their publicly available code on the same training set as the rest of the methods. We also use the ground-truth programs as in TMN. MDETR was already trained on the same training set as the rest of the methods, and hence, we use the publicly available trained model. Also, we report the performance of NS-VQA (Yi et al., 2018) from its original paper and Akula et al. (2021).

## 5 RESULTS AND ANALYSIS

In this section, we first report the systematic generalization performance to novel combinations of visual attributes or linguistic constructs with synthetic images (CLEVR-CoGenT and CLOSURE) and complex natural images (GQA-SGL). Then, we analyze the effect of module composition, module specialization, and pre-training.

### 5.1 SYSTEMATIC GENERALIZATION PERFORMANCE

Tables 1 and 2 show the mean and standard deviation accuracy of TMNs and the rest of approaches for both in-distribution and systematic generalization (see Appendix B, C, D and J for further details). In the following, we first introduce the results in the three datasets, and then, we discuss them:

**CLEVR-CoGenT.** It evaluates the systematic generalization performance to novel combinations of known visual attributes. All models are trained on CLEVR-CoGenT condition A. Our results show that both standard Transformers and TMNs largely outperform the state-of-the-art modular approaches (Vector-NMN and NS-VQA). TMNs' systematic generalization performance also surpasses MDETR, which is the state-of-the-art Transformer-based model proposed to capture long-tail visual concepts. TMNs achieve superior performance over the standard Transformer but their performances are slightly lower than the Transformer with programs (Transformer w/ PR). As expected, Transformer w/ PR outperforms standard Transformer as the use of program facilitates visual grounding.

**CLOSURE.** It evaluates the systematic generalization performance to novel combinations of known linguistic constructs. All models are trained on CLEVR. Our results show that standard Transformers struggle with novel compositions of known linguistic constructs even when the program is provided (Transformer w/ PR). TMNs achieve much better performance than standard Transformers and MDETR, and also outperform the other modular approaches (Vector-NMN and NS-VQA). Remarkably, TMN-Tree improves systematic generalization accuracy over standard Transformers more than $30\%$ in this dataset. Tree structure seems effective for CLOSURE because the questions often have tree structure.

Table 1: *Results on systematic generalization to novel combinations of known visual attributes and linguistic constructs.* Mean and standard deviation of accuracy (%) across at least three repetitions when possible (only one trained model is provided in NS-VQA and MDETR). All methods are tested on in-distribution (CLEVR-CoGenT validation condition A and CLEVR validation) and systematic generalization (CLEVR-CoGenT validation condition B and CLOSURE, indicated in yellow). We trained and test Vector-NMN on CLEVR-CoGenT in our environment while its performances on CLEVR and CLOSURE are cited from the original paper (Bahdanau et al., 2020). "Overall" at the right end is the mean value of accuracies on CoGenT-B and CLOSURE.

| Methods | Visual attributes | | Linguistic constructs | | Overall |
|---|---|---|---|---|---|
| | CoGenT-A (In-Dist.) | CoGenT-B (Syst. Gen.) | CLEVR (In-Dist.) | CLOSURE (Syst. Gen.) | (Syst. Gen.) |
| Transformer | $97.5 \pm 0.18$ | $78.9 \pm 0.80$ | $97.4 \pm 0.23$ | $57.4 \pm 1.6$ | 68.2 |
| Transformer w/ PR | $97.4 \pm 0.56$ | $\mathbf{81.7} \pm 1.1$ | $97.1 \pm 0.14$ | $64.5 \pm 2.5$ | 73.1 |
| TMN-Stack (ours) | $97.9 \pm 0.03$ | $80.6 \pm 0.21$ | $98.0 \pm 0.03$ | $90.9 \pm 0.49$ | 85.3 |
| TMN-Tree (ours) | $98.0 \pm 0.02$ | $80.1 \pm 0.72$ | $97.9 \pm 0.01$ | $\mathbf{95.4} \pm 0.20$ | $\mathbf{87.8}$ |
| Vector-NMN | $98.0 \pm 0.2$ | $73.2 \pm 0.2$ | $98.0 \pm 0.07$ | 94.4 | 83.8 |
| NS-VQA | 99.8 | 63.9 | 99.8 | 76.4 | 70.2 |
| MDETR | 99.7 | 76.2 | 99.7 | 53.3 | 64.8 |

Table 2: *Results on an application to complex natural images with novel combinations of known linguistic constructs.* Mean and standard deviation of accuracy (%) on GQA test-dev and GQA-SGL across at least three repetitions. We use a pre-trained object detector as a feature extractor. The numbers in ( ) denote the mean accuracy on only four question types contained in GQA-SGL (verify, query, choose, and logical). Systematic generalization performance is indicated in yellow.

| Methods | GQA (In-Distribution) | GQA-SGL (Systematic generalization) |
|---|---|---|
| Transformer | $54.9 \pm 0.004$ (67.6) | $47.7 \pm 2.1$ |
| Transformer w/ PR | $54.7 \pm 0.22$ (67.0) | $52.2 \pm 3.2$ |
| TMN-Stack (ours) | $52.8 \pm 0.10$ (64.7) | $50.7 \pm 0.94$ |
| TMN-Tree (ours) | $53.5 \pm 0.24$ (65.2) | $\mathbf{53.7} \pm 1.7$ |
| MMN | $55.6 \pm 0.12$ (67.8) | $51.0 \pm 1.5$ |

**GQA-SGL.** We evaluate the systematic generalization performance to novel combinations of known linguistic constructs with natural images. All models are trained on GQA. In Table 2, we evaluate the exact matching between the predicted and ground-truth answers, on both GQA and GQA-SGL. TMN-Tree achieves superior performance on systematic generalization with a smaller drop of in-distribution accuracy compared to standard Transformers. Although MMN is the recently proposed NMN for GQA and achieves great in-distribution performance, TMN-Tree outperforms MMN on systematic generalization.

**Discussion of the Results.** Our results in CLOSURE and GQA-SGL show that TMNs (i.e., introducing modularity to Transformers) improve the systematic generalization performance to novel combinations of known linguistic constructs (i.e., novel compositions of sub-tasks) of Transformers and NMNs. The performance gain in CLOSURE is relatively larger than that in GQA-SGL tests. One reason to explain this is the complexity of the questions: the maximum program length is 26 in CLOSURE, while it is 9 in GQA. CLOSURE provides more complex questions which require stronger systematic generalization capabilities. Results in CLEVR-CoGenT show that TMNs do not improve systematic generalization of Transformers for novel compositions of visual attributes. This may be expected as D'Amario et al. (2021) demonstrated that the visual feature extractor is the key component for generalization to novel combinations of visual attributes, and Transformer and TMNs have the same visual feature extractor. Finally, we observe that TMNs achieve the best accuracy across all tested methods in CLOSURE and GQA-SGL, and competitive accuracy with the best performing method in CLEVR-CoGenT.

Table 3: *Effect of specialization of modules.* Mean and standard deviation of accuracy (%) on CLEVR and CLOSURE across three repetitions with different libraries of modules. "Individual" denotes that a module is assigned to a single sub-task. "Semantic group" and "Random group" denote that a module is assigned to a group including sub-tasks that are semantically similar or randomly selected, respectively. "Order" denotes that a module is selected by the order of a sub-task in a program.

| Library of modules | The number of modules | Specialization of modules | CLEVR (In-Distribution) | CLOSURE (Syst. Generalization) |
|---|---|---|---|---|
| Individual | 26 | ✓ | $98.0 \pm 0.030$ | $90.9 \pm 0.49$ |
| Semantic group | 12 | ✓ | $98.1 \pm 0.014$ | $\mathbf{93.7 \pm 0.55}$ |
| Random group | 12 | ✓ | $97.9 \pm 0.059$ | $93.0 \pm 0.29$ |
| Order | 12 | | $92.2 \pm 1.5$ | $68.4 \pm 1.8$ |

Table 4: *Ablation analysis.* Mean and standard deviation of accuracy (%) on CLEVR and CLOSURE across three repetitions. We add imitations of properties of TMN, i.e., Variable number of layers (VL) and Split tokens (ST) to the standard Transformer w/ PR. SoM stands for the specialization of the modules.

| Methods | Properties | | | CLEVR (In-Distribution) | CLOSURE (Syst. Generalization) |
|---|---|---|---|---|---|
| | VL | ST | SoM | | |
| Transformer w/ PR | | | | $97.1 \pm 0.14$ | $64.5 \pm 2.5$ |
| Transformer w/ PR + VL | ✓ | | | $94.0 \pm 1.8$ | $60.2 \pm 2.9$ |
| Transformer w/ PR + VL + ST | ✓ | ✓ | | $56.8 \pm 0.042$ | $43.6 \pm 0.59$ |
| TMN-Stack | ✓ | ✓ | ✓ | $98.0 \pm 0.030$ | $\mathbf{90.9 \pm 0.49}$ |

## 5.2 ANALYSIS

**Behavioral Analysis.** We analyze the behavior of our trained modules, i.e., how Transformer modules work, by visualizing attention maps in Appendix H. The qualitative results indicate that the modules learn their functions properly. We also analyze the Transformer w/ PR in Appendix I.

**Module Specialization.** While the question-specific composition of modules is a key property of NMNs, a recent work reports that a degree of modularity of the NMN also has large influence on systematic generalization (D'Amario et al., 2021). Here we also investigate this aspect for TMNs. Table 3 shows the mean accuracy of TMN-Stack with four different libraries of modules, each with a different degree of specialization. "Individual" denotes that each module is assigned to a single sub-task (26 sub-tasks defined in CLEVR in total). This is the library used in the previous sections. "Semantic group" denotes that each module tackles a group of sub-tasks that are semantically similar. "Random group" denotes that each module tackles a group of sub-tasks randomly selected. "Order" denotes that each module is always placed in the same position of the question-specific network, independently of the sub-task performed in that position. If there are more sub-tasks in the program than modules in the library, the modules are repeated following the same order. In this way, we compare the importance of modules tackling sub-tasks with the importance of having a question-specific network. See Appendix E for further details about each library.

Results shows that the systematic generalization accuracy largely drops when the modules are not specialized, even if the number of modules is the same (compare "Order" with "Semantic group" and "Random group"). The degree of modularity (Individual vs Semantic group) and how sub-tasks are grouped (Semantic vs Random) also affects the systematic generalization performance. These results suggest that question-specific composition of modules is not enough and specialization of modules is crucially important. Finally, note that the results of Semantic group and Random group also show that TMN performs better than the standard Transformer ($93.7$ vs $64.5$) even if the total number of parameters that all modules have is the same as the number of parameters of the standard Transformer (i.e., TMN has $12$ modules of one layer each, while Transformers have $12$ layers).

**Ablation Analysis.** We investigate the contribution of TMN's specialization of modules (SoM) to TMNs' superior performance over standard Transformer. Besides SoM, TMNs also have a variable

Table 5: *Effect of pre-training.* Mean accuracy (%) on CLEVR-CoGenT validation condition B and CLOSURE with two different feature extractors, which are an object detector pre-trained on Visual Genome (VG) and a ResNet-101 (without pre-training on VG).

| Methods | CLEVR-CoGenT | | | CLOSURE | | |
|---|---|---|---|---|---|---|
| | w/o VG | w/ VG | $\Delta$ | w/o VG | w/ VG | $\Delta$ |
| Transformer | 78.9 | 87.5 | 8.6 | 57.4 | 62.1 | 4.7 |
| Transformer w/ PR | **81.7** | **88.7** | 6.9 | 64.5 | 64.5 | 0.005 |
| TMN-Stack (ours) | 80.6 | 86.9 | 6.3 | 90.9 | 92.7 | 1.7 |
| TMN-Tree (ours) | 80.1 | 86.4 | 6.4 | **95.4** | **96.4** | 1.0 |

number of layers, while standard Transformer have a fixed number of layers. We analyze if it is this difference in number of layers or the SoM that is causing TMNs' higher accuracy. To do so, we add a property named Variable number of layers (VL) to the Transformer w/ PR. With the VL, the Transformer w/ PR uses as many encoder layers as the program length (e.g., 18 layers are used to output an answer for an input program with a length of 18). VL can be regarded as an imitation of the question-specific compositions of the modules in TMN, but each layer is not specialized for specific sub-tasks. Another key difference between TMNs and Transformers besides SoM is that TMN modules takes as argument tokens related to the sub-task as indicated by the program, while every layer of standard Transformer takes all input tokens in the program, because Transformers' layers are not specialized to sub-tasks. Splitting the program tokens may help the model to steer the computations that should be executed at each layer. To investigate this, we add this property, named as Split Tokens (ST), to the Transformer w/ PR.

Table 4 shows that any of the aforementioned properties do not improve the systematic generalization performance without the specialization of modules to sub-tasks, i.e., SoM. Even with Split tokens, the performance does not improve because the layers can not be specialized and splitting the token can spoil the flexibly of the standard Transformer.

**Pre-training.** We analyze how the additional image data from other domains affects systematic generalization to novel combinations of visual attributes or linguistic constructs, i.e., CLEVR-CoGenT and CLOSURE, respectively. Table 5 shows the mean accuracy with two different visual features, namely, regional features extracted by an object detector pre-trained on Visual Genome (VG) and grid features extracted by a ResNet-101 without pre-training on VG. Across all the tested models, we observe, as expected, that the performance gains from pre-training the visual feature extractor are larger for generalization to novel visual combinations and smaller for novel linguistic combinations. This highlights the need to control pre-training in order to ensure a fair comparison of different methods for systematic generalization. See Appendix F and G for further experiments.

## 6 CONCLUSIONS

We have compared the systematic generalization capabilities of TMNs with the most promising approaches in the literature, i.e., Transformers and NMNs, on three VQA datasets. TMNs achieve state-of-the-art systematic generalization performance across all tested benchmarks. We have shown that the question-specific composition of Transformer modules and the specialization of the modules to each sub-task are the key of the performance gains. Also, our results highlight that a fair comparison for systematic generalization requires controlling that all compared methods are pre-trained in the same way. We hope that our investigation unleash the potential of modular approaches for systematic generalization as they motivate promising future work towards creating large-scale modular architectures.

**Limitations.** TMNs have similar limitations as NMNs, i.e., it is challenging to decompose a question into sub-tasks. We have used the sub-tasks and the programs provided in the datasets generated by humans. This makes it difficult to apply NMNs and TMNs to other datasets that do not contain the programs. Another limitation is that we only investigated two types of systematic generalization (novel compositions of visual attributes and linguistic constructs). These limitations are key open questions that we will tackle in future work.

**Reproducibility Statement.** The source code and new dataset used in this study are included in the supplementary materials for review and will be made publicly available on GitHub.com upon acceptance. URLs of other existing assets (datasets, source code and trained models) used in this study are provided in the reference section. Experimental setup is described in Section 4.

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

# APPENDIX

## A GQA-SGL DATASET

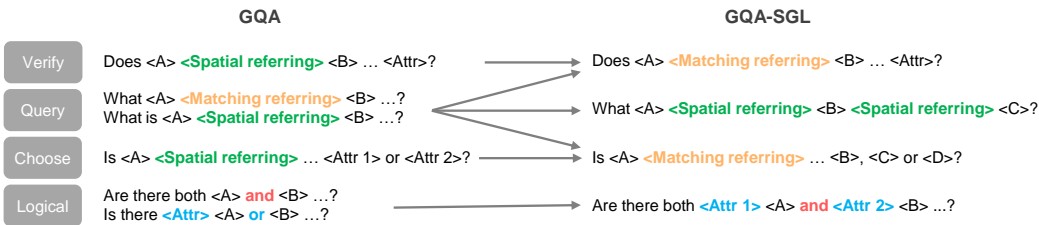

Figure 4: *GQA-SGL test dataset for systematic generalization with natural images.* GQA-SGL is a novel test set we built based on GQA to evaluate systematic generalization performance. We generate new questions with ground-truth programs that belong to four question types (verify, query, choose, and logical) by combining the existing programs. <A,...,D> and <Attr> denote an object (e.g., chair) and an attribute (e.g., blue). <Spatial referring> and <Matching referring> are a referring expression such as " is right of " and " is the same color as ", respectively.

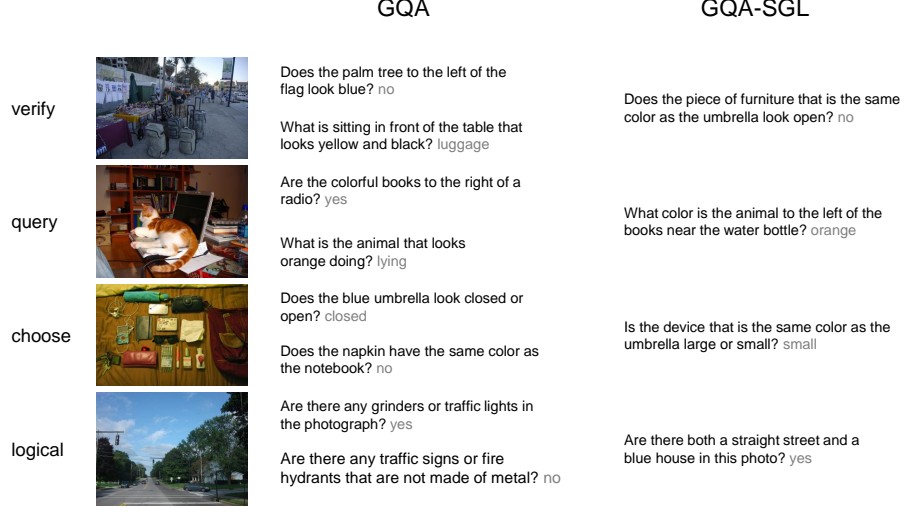

Figure 5: Examples of test questions from GQA and GQA-SGL for the same images.

We generate GQA-SGL, a novel GQA split to test Systematic Generalization to the novel Linguistic combinations of models trained on GQA. GQA-SGL contains 200 new questions with their ground-truth programs. 97.4% of the original GQA questions can be divided into four question types: verify, query, choose, and logical. Fig. 4 describes the GQA-SGL data generation process. On the left, we first list out all ground-truth programs that belong these question types. The arguments in parenthesis <·> denote either objects (*e.g.*, A and B), attributes (*e.g.*, Attr), or referring expression. Those can be *spatial referring*, where an object or group of objects is selected based on the relation with another object or group of objects (*e.g., the object A which is left to objects B*); *matching referring* where an objects/group is selected depending on some matching properties with another object/group; *and* or *or* where either or one between two objects/groups of objects are selected.

On the right of Fig. 4, we generate new ground-truth programs by combining attributes and referring expressions which never appeared in GQA. We emphasize the systematic shift between GQA and GQA-SGL linguistic construct by highlighting referring expressions and attributes through different colors.

We generate 50 questions for each question type as shown in Fig. 5.

Table 6: *Supplemental results on CLEVR-CoGenT.* Mean and standard deviation of accuracy (%) for each question type in CLEVR-CoGenT Condition B across at least three repetitions.

| Methods | Exist | Count |
|---|---|---|
| Transformer | $87.8 \pm 0.69$ | $73.3 \pm 0.13$ |
| Transformer w/ PR | **90.4** $\pm 0.90$ | **76.8** $\pm 0.99$ |
| TMN-Stack (ours) | $88.4 \pm 0.069$ | $75.1 \pm 0.27$ |
| TMN-Tree (ours) | $87.8 \pm 0.62$ | $74.3 \pm 0.83$ |
| Vector-NMN | $84.4 \pm 0.35$ | $70.4 \pm 0.44$ |
| MDETR | 85.4 | 75.2 |

| Methods | Compare Integer | Compare Attribute | Query Attribute |
|---|---|---|---|
| Transformer | $88.6 \pm 0.41$ | $85.8 \pm 0.68$ | $73.2 \pm 1.7$ |
| Transformer w/ PR | **89.8** $\pm 0.45$ | **88.3** $\pm 1.1$ | $76.3 \pm 1.6$ |
| TMN-Stack (ours) | $88.1 \pm 0.20$ | $86.3 \pm 0.22$ | **76.6** $\pm 0.28$ |
| TMN-Tree (ours) | $89.0 \pm 0.66$ | $87.8 \pm 0.69$ | $75.3 \pm 0.75$ |
| Vector-NMN | $81.3 \pm 0.52$ | $78.1 \pm 0.79$ | $66.5 \pm 1.05$ |
| MDETR | 81.3 | 82.6 | 68.9 |

Table 7: *Supplemental results on CLOSURE.* Mean and standard deviation of accuracy (%) on each CLOSURE test across at least three repetitions.

| Methods | and_mat_spa | or_mat | or_mat_spa |
|---|---|---|---|
| Transformer | $77.0 \pm 5.2$ | $19.1 \pm 0.55$ | $25 \pm 11$ |
| Transformer w/ PR | $77.6 \pm 6.9$ | $39.7 \pm 1.9$ | $36.4 \pm 5.8$ |
| TMN-Stack (ours) | $98.4 \pm 0.19$ | $77.8 \pm 1.5$ | $72.7 \pm 2.6$ |
| TMN-Tree (ours) | **98.5** $\pm 0.03$ | $89.5 \pm 0.62$ | **90.1** $\pm 0.8$ |
| Vector-NMN | $86.3 \pm 2.5$ | **91.5** $\pm 0.77$ | $88.6 \pm 1.2$ |
| MDETR | 8.63 | 34.1 | 18.4 |

| Methods | embed_spa_mat | embed_mat_spa | compare_mat | compare_mat_spa |
|---|---|---|---|---|
| Transformer | $97.0 \pm 1.3$ | $55.2 \pm 4.4$ | $65.7 \pm 2.8$ | $62.7 \pm 2.3$ |
| Transformer w/ PR | $59.7 \pm 2.8$ | $78.1 \pm 5.0$ | $85.7 \pm 4.4$ | $74.4 \pm 6.6$ |
| TMN-Stack (ours) | **98.8** $\pm 0.00$ | $92.7 \pm 0.84$ | $77.8 \pm 1.5$ | $72.7 \pm 2.6$ |
| TMN-Tree (ours) | $98.7 \pm 0.16$ | $92.8 \pm 0.42$ | **99.0** $\pm 0.03$ | **99.3** $\pm 0.11$ |
| Vector-NMN | $98.5 \pm 0.13$ | **98.7** $\pm 0.19$ | $98.5 \pm 0.17$ | $98.4 \pm 0.3$ |
| MDETR | 99.3 | 77.4 | 69.3 | 66.0 |

Table 8: *Supplemental results on GQA-SGL.* Mean and standard deviation of accuracy (%) for four question types in GQA-SGL across at least three repetitions. We use a pre-trained object detector as a feature extractor.

| Methods | Four question types in GQA-SGL | | | |
|---|---|---|---|---|
| | verify | query | choose | logical |
| Transformer | $50.0 \pm 1.6$ | $35.3 \pm 5.0$ | $50.0 \pm 7.5$ | $55.3 \pm 4.1$ |
| Transformer w/ PR | $51.6 \pm 7.7$ | $45.6 \pm 5.6$ | **56.0** $\pm 6.6$ | $55.6 \pm 5.4$ |
| TMN-Stack (ours) | $51.6 \pm 4.1$ | **49.6** $\pm 5.0$ | $44.8 \pm 3.7$ | $56.4 \pm 7.1$ |
| TMN-Tree (ours) | **56.0** $\pm 7.3$ | $48.0 \pm 5.5$ | $47.2 \pm 3.2$ | **64.0** $\pm 4.6$ |
| MMN | $46.7 \pm 4.7$ | $50.7 \pm 5.0$ | $47.3 \pm 1.9$ | $59.3 \pm 3.4$ |

## B SUPPLEMENTAL RESULTS ON CLEVR-COGENT

We report the systematic generalization performance for each question type on CLEVR-CoGenT in Table 6. All methods are trained on CLEVR-CoGenT condition A and tested on CLEVR-CoGenT validation condition B.

## C SUPPLEMENTAL RESULTS ON CLOSURE

We report the systematic generalization performance for each test in CLOSURE in Table 7. All methods are trained on CLEVR and tested on CLOSURE tests.

## D SUPPLEMENTAL RESULTS ON GQA-SGL

We report the systematic generalization performance for each question type on GQA-SGL in Table 8. All methods are trained on GQA balanced split.

## E LIBRARIES OF MODULES

We design three libraries of modules for TMN, "Individual", "Semantic group", and "Random group", as shown in Table 9. Unlike these libraries, a module in "Order" library is selected by the order of a sub-task in a given program and thus in this library the modules are not specialized for specific sub-tasks.

## F EFFECT OF REGIONAL FEATURES

We compare the systematic generalization performances on CLEVR-CoGenT with grid features and regional features, as shown in Table 10. We use ResNet-101 and region of interest pooling to extract regional features, while we use only the same ResNet-101 to extract grid features. In case of the standard Transformer and our TMNs, the performance gaps are less than $1.5\%$. In case of the standard Transformer with ground-truth programs, the performance gap is slightly larger than it $(3.6\%)$.

## G EFFECT OF PRE-TRAINING WITH IMAGE–TEXT PAIRS

Table 11 reports the results of TMN-Tree and state-of-the-art Transformer-based models (MDETR (Kamath et al., 2021) and LXMERT (Tan & Bansal, 2019)) on GQA and GQA-SGL. The table also indicates the pre-training datasets with image-text pairs. The methods use different pre-training datasets, and hence, they cannot be compared in a fair manner. Yet, we observe that TMN-Tree strike a balance between amount of pre-training data for the Transformer encoders and systematic generalization performance.

Table 9: *Libraries of modules.* "Individual" library has 26 modules, while "Semantic group" and "Random group" libraries have 12 modules.

| Module number | Individual | Semantic group | Random group |
|---|---|---|---|
| 0 | scene | scene | query_color |
| 1 | count | count | query_shape |
| 2 | exist | exist | union |
| 3 | intersect | intersect | same_shape |
| 4 | relate | relate | equal_shape |
| 5 | union | union | greater_than |
| 6 | unique | unique | equal_material |
| 7 | greater_than | greater_than
less_than
equal_integer | filter_shape
intersect
unique |
| 8 | less_than | equal_color
equal_material
equal_shape
equal_size | same_material
filter_material
count
same_size |
| 9 | equal_color | filter_color
filter_material
filter_shape
filter_size | equal_size
query_material
scene
equal_color |
| 10 | equal_integer | query_color
query_material
query_shape
query_size | query_size
less_than
exist
relate |
| 11 | equal_material | same_color
same_material
same_shape
same_size | filter_size
equal_integer
filter_color
same_color |
| 12 | equal_shape | | |
| 13 | equal_size | | |
| 14 | filter_color | | |
| 15 | filter_material | | |
| 16 | filter_shape | | |
| 17 | filter_size | | |
| 18 | query_color | | |
| 19 | query_material | | |
| 20 | query_shape | | |
| 21 | query_size | | |
| 22 | same_color | | |
| 23 | same_material | | |
| 24 | same_shape | | |
| 25 | same_size | | |

Table 10: *Effect of regional features.* Mean and standard deviation of accuracy (%) on CLEVR-CoGenT validation condition B with grid features and regional features.

| Methods | Grid features | Regional features |
|---|---|---|
| Transformer | $78.9 \pm 0.80$ | $77.4 \pm 0.68$ |
| Transformer w/ PR | $\mathbf{81.7} \pm 1.1$ | $78.2 \pm 0.44$ |
| TMN-Stack (ours) | $80.6 \pm 0.21$ | $79.4 \pm 0.37$ |
| TMN-Tree (ours) | $80.1 \pm 0.72$ | $\mathbf{80.9} \pm 0.25$ |

Table 11: *Comparison with pre-trained state-of-the-art VQA methods.* Mean accuracy (%) on four question types (verify, query, choose, and logical) in GQA test-dev and GQA-SGL. The pre-training refers to datasets with image-text pairs used during training besides GQA. All methods except TMN-Tree are pre-trained with a large amount of image-text pairs. For the methods with †, we use official trained models. VG, COCO, Fk, GQAu and VQA denote Visual Genome, MS COCO, Flickr30k, GQA unbalanced split, and VQA-v2, respectively.

| Methods | Pre-training data | GQA | GQA-SGL |
|---|---|---|---|
| TMN-Tree (ours) | - | 65.2 | 53.7 |
| MDETR † | VG, COCO, Fk, GQAu | 73.9 | 59.0 |
| LXMERT † | VG, COCO, VQA | 68.9 | 58.0 |

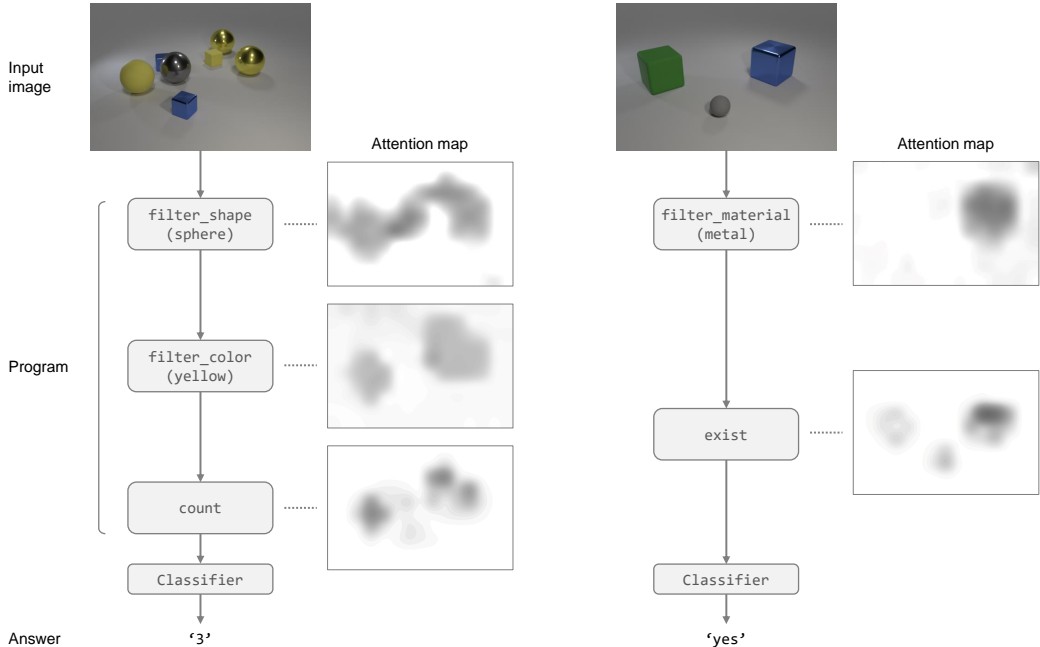

Figure 6: *Visualization of TMN's behavior with examples in CLEVR.* The left program is {scene, filter_shape(sphere), filter_color(yellow), count} and the right program is {scene, filter_material(metal), exist}. Attention maps of modules corresponding to the sub-tasks are shown in grey scale.

## H  BEHAVIOURAL ANALYSIS OF MODULES

Figure 6 shows an visualization of how Transformer modules work, obtained by using examples in CLEVR, i.e., in-distribution case. In the left example, we give TMN-Stack an input image with a program {scene, filter_shape(sphere), filter_color(yellow), count}. Three modules corresponding to the program operate their sub-tasks sequentially and then a classifier outputs an answer. We visualize the attention map in each module, where darker grey represents higher attention scores. We average attention scores across all attention heads. We can see that filter_shape(sphere) and filter_color(yellow) modules highly attend to regions with sphere and yellow objects, respectively. More precisely, the queries of the regions highly attend to the keys of sphere and yellow which are the input for each module, and the representations of the regions change by adding the values of sphere and yellow. Although filter_color(yellow) module executes its sub-task after filter_shape(sphere), filter_color(yellow) also attends to a yellow cube as well as yellow spheres. In the right example, we give an another image with a program {scene, filter_material(metal), exist}. These results indicate that the modules learn their

Table 12: *Effect of model size on Transformers.* We compare Transformers and Transformers w/ PR with the different number of encoder layers.

| Methods | The number of layers | CLEVR (In-Distribution) | CLOSURE (Syst. Generalization) |
|---|---|---|---|
| Transformer | 12 | $97.4 \pm 0.23$ | $57.4 \pm 1.6$ |
| | 6 | $96.8 \pm 0.0068$ | $\mathbf{60.6 \pm 3.8}$ |
| | 3 | $91.8 \pm 0.40$ | $56.1 \pm 2.0$ |
| Transformer w/ PR | 12 | $97.1 \pm 0.14$ | $\mathbf{64.5 \pm 2.5}$ |
| | 6 | $95.7 \pm 0.40$ | $63.5 \pm 2.7$ |

functions properly. Figure 7 shows a behavioural analysis on an example in CLOSURE, i.e., novel combinations of known linguistic constructs. In this example, we give TMN-Tree an input image with a program {`scene`, `filter_shape(cube)`, `scene`, `filter_size(large)`, `filter_shape(cube)`, `unique`, `same_material`, `filter_shape(sphere)`, `union`, `count`}. `scene` represents the beginning of a thread and this program contains two threads. TMN-Tree operates the threads individually and merges them with `union` (`OR`). For simplicity, we overlay attention maps on the input image. The result show that the each module works properly as we expect even though this example is much more complex than the ones shown in Fig. 6 and also the sequence of the sub-tasks are unseen in the training.

## I BEHAVIOURAL ANALYSIS OF TRANSFORMER W/ PR

Figure 8 and 9 show how Transformer w/ PR works on the same example in CLOSURE shown in Fig. 7 and an example in CLEVR similar to the CLOSURE example. Both examples contain `union` (`OR`), however the CLEVR example is in-distribution while the CLOSURE example requires the model to handle novel combinations of known linguistic constructs, i.e., systematic generalization. We visualize the attention map of the head token for the visual regions and the input sequence of the tokens. The results show that the Transformer w/ PR first attends to the input program then changes where to attend in the visual regions.

In Fig. 9, i.e., in-distribution case, the model attends to the token almost from left to right (i.e., order of sub-tasks) and after reaching the last token it stop to attend the program. Specifically, the model first attends to the regions with metal cube from layer 2 to 3 and the regions of sphere at layer 5, then attends to the both regions at layer 6 after attending the `union` at the previous layer. Finally, the model attends to the `count`. This result indicates that the Transformer w/ PR learns how to operate the input programs to output the correct answers.

In Fig. 8, i.e., systematic generalization case, the input program contains `same_material` in addition to `union`. Both sub-tasks appear in the training while the combination is novel. Fig. 8 shows that the model tried to operate the input program in the same manner as the CLEVR example shown in Fig. 9 but failed to lead a correct answer. Specifically, at the same layer as the CLEVR example, the model attends to the `union` and `count`, however the attended regions are not correct. After that the model seems to operate `same_material` that should be operated before `union`. These results indicate that Transformer w/ PR struggles to operate the programs that contain the novel combinations of the sub-tasks.

## J ABLATION STUDIES FOR TRANSFORMERS

We test Transformers and Transformers w/ PR with the different number of encoder layers to investigate the effect of model size. We train all models in the same way we described in the main paper. Table 12 shows that the performance of Transformers with fewer layers slightly drops except Transformer with 6 layers. These results show that we cannot improve the systematic performance by simply changing the number of layers.

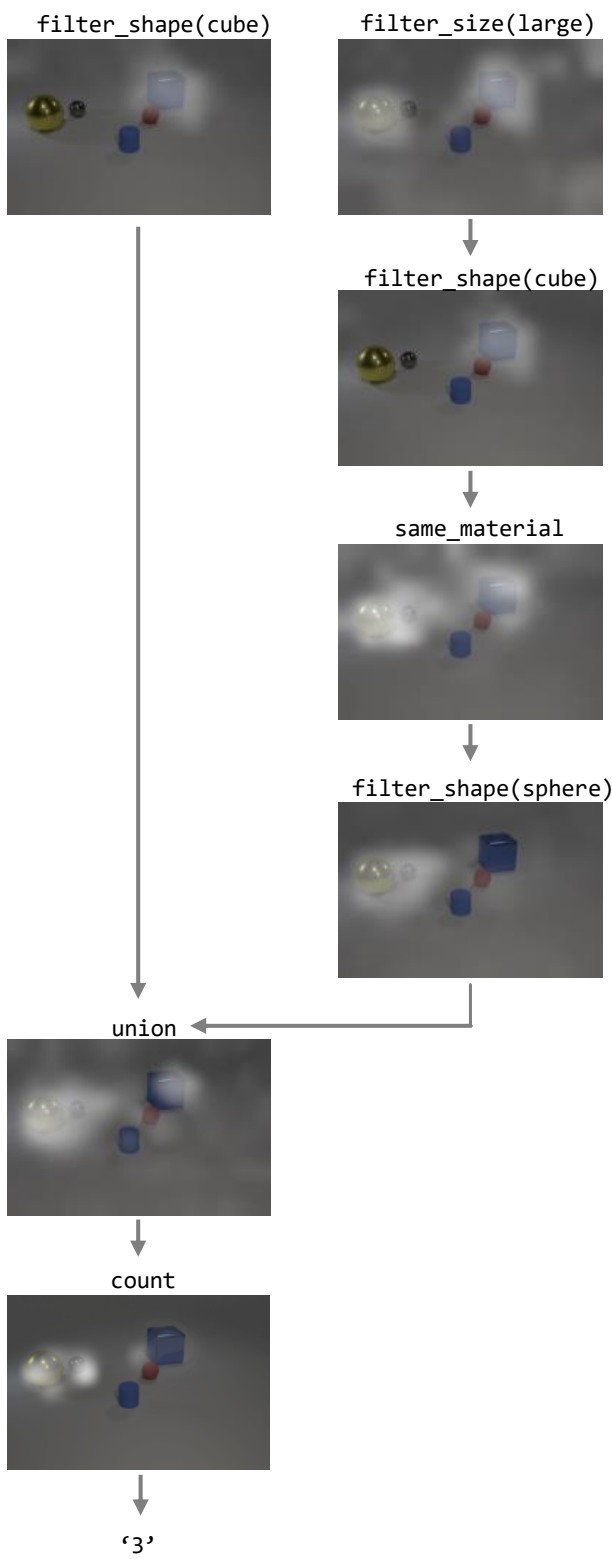

Figure 7: *Visualization of TMN-Tree's behavior with an example in CLOSURE.* The program is {scene, filter_shape(cube), scene, filter_size(large), filter_shape(cube), unique, same_material, filter_shape(sphere), union, count}. Attention maps of modules corresponding to the some sub-tasks are shown over the input image.

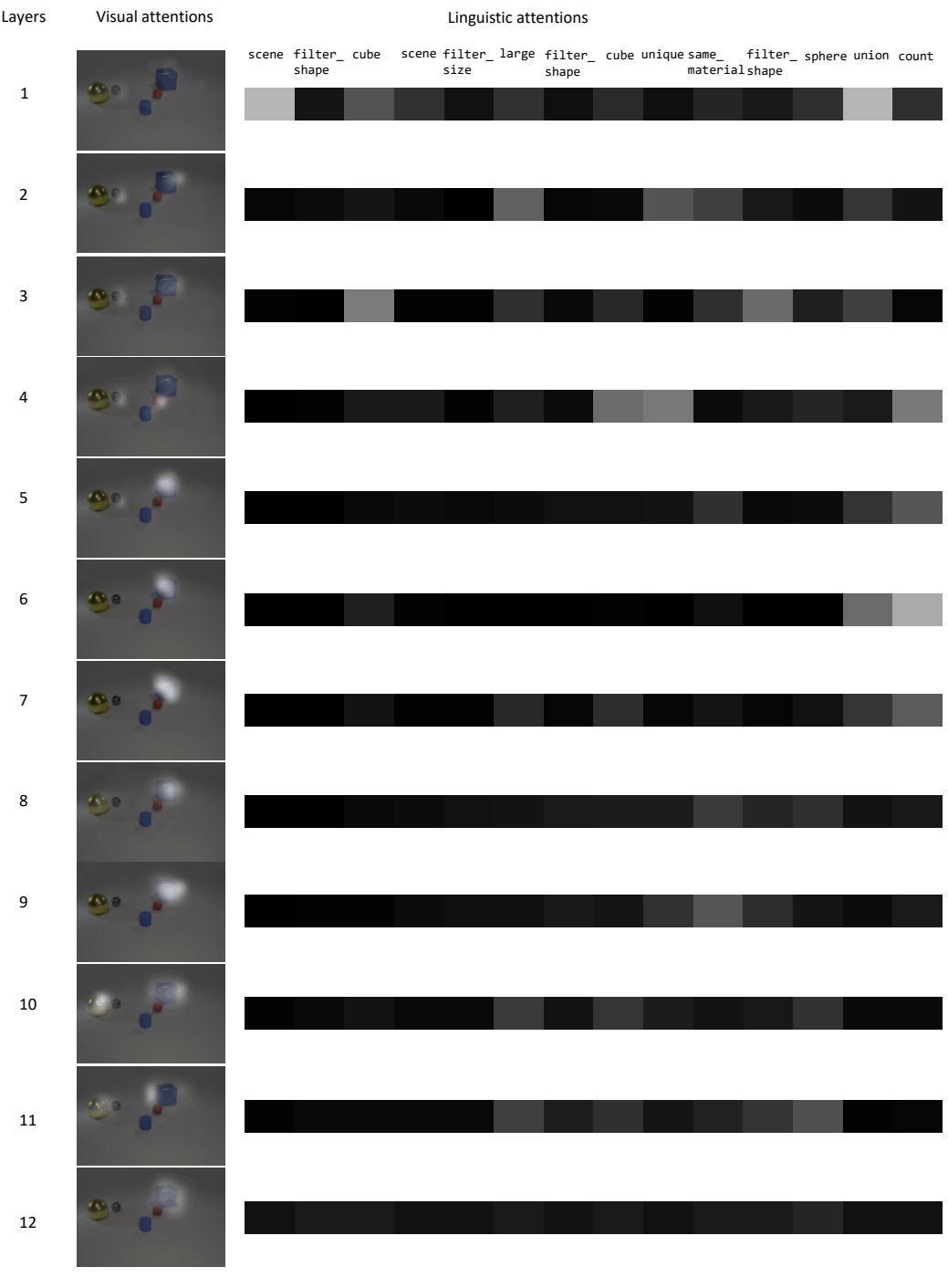

Figure 8: *Visualization of behavior of Transformer w/ PR with an example in CLO-SURE.* The program is {scene, filter_shape(cube), scene, filter_size(large), filter_shape(cube), unique, same_material, filter_shape(sphere), union, count}.

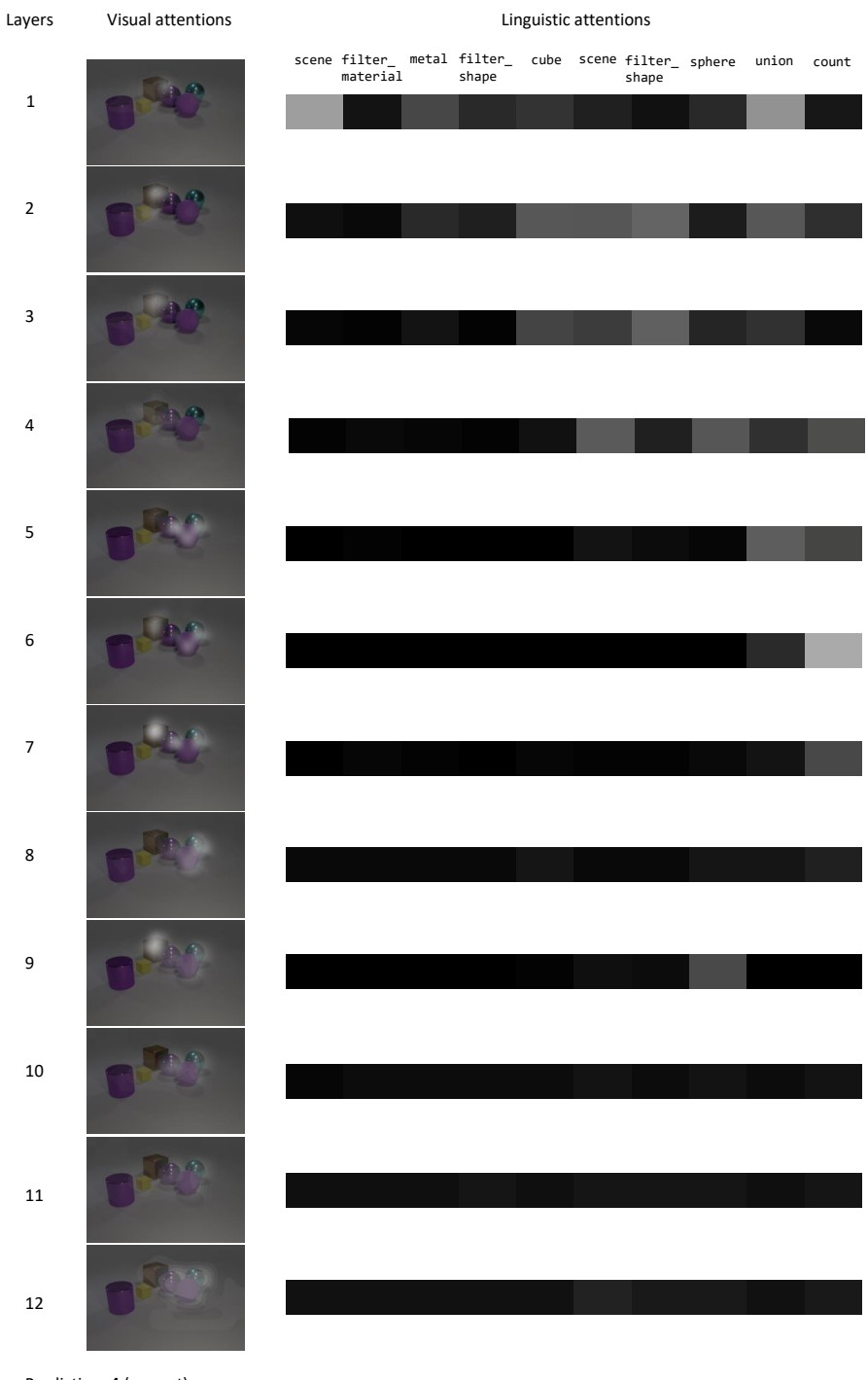

Figure 9: *Visualization of behavior of Transformer w/ PR with an example in CLEVR. The program is* {scene, filter_material(metal), filter_shape(cube), scene, filter_shape(sphere), union, count}.

