# OpenReview forum: "Transformer Module Networks for Systematic Generalization in Visual Question Answering"
_ICLR.cc/2023/Conference — Submitted to ICLR 2023_

### Official Review · Reviewer_zXs6 · 2022-10-24

**Confidence:** 2
**Correctness:** 3
**Technical Novelty And Significance:** 2
**Empirical Novelty And Significance:** 3
**Recommendation:** 5

**Clarity, Quality, Novelty And Reproducibility:**

This paper is technically sound, with sufficient clarity. The details provided are fully explained and can be reproducible. However, it appears to be limited in scientific contribution.

**Strength And Weaknesses:**

•	Strengths:

1.	The proposed method is technically sound and easy to understand. It is reasonable to have a model that makes use of the known advantages of Transformers and module networks.

2.	It shows better performance when compared against existing methods.

•	Weaknesses:

1.	While the proposed model is straightforward and easy to understand, it appears that the model brings very little insight into the systematic generalization task itself. It is well studied that module networks are a potential approach for systematic generalization but it is extremely brittle and ad-hoc which hinders its application in wider settings. Meanwhile, models with better representation learning such as Transformers are beneficial for all machine learning tasks, not just for systematic generalization. The proposed method does not offer any fundamental contributions toward solving systematic generalization.

2.	Experiments with original transformers do not sound reasonable as the original design of transformer is for translation tasks with having access to a more generous amount of training data. There is no reason to have to use 12 layers for transformers when applying it to a custom problem. The poor performance of transformers in the experiments might simply be due to overfitting.


**Summary Of The Paper:**

This paper addresses the problem of systematic generalization in VQA. It proposes a new model that takes advantage of the learning capability of Transformers and the compositional modeling of Module Networks. The proposed model achieves state-of-the-art performance on three different VQA datasets including standard testbeds for VQA systematic generalization.

**Summary Of The Review:**

The paper is well written and easy to follow. However, it is limited in contribution to advance the systematic generalization task.

---

### Official Review · Reviewer_pauq · 2022-10-24

**Confidence:** 4
**Correctness:** 3
**Technical Novelty And Significance:** 2
**Empirical Novelty And Significance:** 3
**Recommendation:** 5

**Clarity, Quality, Novelty And Reproducibility:**

The paper is well-organized and easy to understand.

The novelty is not enough. The proposed model only replaces the previous neural module with a transformer.

**Strength And Weaknesses:**

Strength:

The paper focuses on the model robustness against VQA bias, which is a common problem in the VQA field.

The paper proposed a new systematic generalization dataset GQA-SGL based on the GQA. The new dataset can be a new challenge for future work in this field.

Weakness:

The novelty is limited. The proposed model is a modular network composed of more expressive transformer modules.

The proposed model also suffers from the weakness of the previous modular network. It needs program supervision and relies on the performance of the question parser. Since all the experiments have a groundtruth program at test time, whether the question parser has the same systematic generalization ability is not verified.

The improvement on the CLEVR-CoGenT and CLOSURE dataset is not consistent. The proposed model performs worse than the baseline on the CLEVR-CoGenT but improves on the CLOSURE. I suppose the CLEVR-CoGenT and CLOSURE have novel images and questions respectively. However, the TMN can ignore the questions by accessing the groundtruth programs.
The authors suggest that the proposed model and baseline share the same visual feature extractor. Thus authors may verify this by training a new extractor from scratch similar to FiLM[1].

[1] Perez E., Strub F., de Vries H., Dumoulin V., Courville, A. FiLM: Visual Reasoning with a General Conditioning Layer. AAAI 2017

**Summary Of The Paper:**

The paper focuses on the systematic generalization visual question answering where the test set contains novel combinations of training concepts. The authors propose a new modular network whose modules are transformers.

**Summary Of The Review:**

The paper proposes a new dataset. But the proposed model is not novel and its effectiveness is not fully verified. Thus I give borderline.

---

### Official Review · Reviewer_o4YA · 2022-10-24

**Confidence:** 5
**Correctness:** 2
**Technical Novelty And Significance:** 2
**Empirical Novelty And Significance:** 2
**Recommendation:** 5

**Clarity, Quality, Novelty And Reproducibility:**

The paper clarified its contributions and approach clearly.
In general, the paper can be reproduced and is of fair quality.
However, it lacks novelty since the tree structure and substituting the modular with the TRM encoder are not novel.

**Strength And Weaknesses:**

*[Strength]*

The motivation that combining the strength of NMN and Transformer is contributing.

*[Weakness]*
1. In the approach, the paper directly substitutes the module in the NMN with the TRM encoder. However, it lacks ablation experiments and discussion about if the TRM structure outperforms all types of module in NMN-based method. (e.g., in n2mn[1] Table 1, the attention based modules like find, relocate, filter modules and the answer based modules like describe, compare and exist modules)
2. The Tree modular modular is not novel in NMN domain, [1] uses similar manner in Figure 3.
3. Lacks of SOA TRM-based and NMN-based model performance comparisons. In addition, the SOA method on GQA dataset is 72.1 from [2], and there is [3] with 69.46 on GQA-dev




[1] Hu, Ronghang, et al. "Learning to reason: End-to-end module networks for visual question answering." Pr oceedings of the IEEE international conference on computer vision. 2017.

[2] Nguyen, Binh X., et al. "Coarse-to-Fine Reasoning for Visual Question Answering." Proceedings of the IEEE/CVF Conference on Computer Vision and Pattern Recognition. 2022.

[3] Kim, Eun-Sol, et al. "Hypergraph attention networks for multimodal learning." Proceedings of the IEEE/CVF conference on computer vision and pattern recognition. 2020.

**Summary Of The Paper:**

This paper mainly focuses on combining the strengths of Transformer and NMN by introducing a novel NMN based on compositions of Transformer modules named Transformer Module Network (TMN). The model is evaluated on CLEVR-CoGenT, CLOSURE, and GQA-SGL.

**Summary Of The Review:**

The motivation that combining the strengths of the two structures is good. However, directly substituting the modular with TRM seems not novel to me. Besides, there are no experiments to prove that the TRM encoder outperforms all previous NMN modular in the task, thus, it does not convince me of its superiority by directly substituting all NMN modulars. In addition, it lacks comparisons of the SOA methods. And from my knowledge, the performance is not SOA (the last point in the weakness section). Overall, I'm leaning to reject it.

---

### Official Review · Reviewer_W6Kf · 2022-10-25

**Confidence:** 3
**Correctness:** 4
**Technical Novelty And Significance:** 4
**Empirical Novelty And Significance:** 4
**Recommendation:** 5

**Clarity, Quality, Novelty And Reproducibility:**

The proposed method is clear and novel. The quality of this paper looks favorable overall.

**Strength And Weaknesses:**

Strength
- The observation about the comparison between the neural module networks and the Transformer models is novel and interesting.
- The authors provide a comprehensive ablation study to make the final model more solid.


Weakness
- It would be better to add qualitative results to see the behavior of the proposed method and the existing methods. Also, it would be better to discuss why the conventional Transformer models perform worse than the neural module networks by analyzing the samples.

**Summary Of The Paper:**

The authors reveal that Neural Module Networks (NMNs), i.e., question-specific compositions of modules that tackle a sub-task, achieve better or similar systematic generalization performance than the conventional Transformers, even though NMNs’ modules are CNN-based.
To address this shortcoming of Transformers with respect to NMNs, in this paper, the authors investigate whether and how modularity can bring benefits to Transformers.
Namely, they introduce a Transformer Module Network (TMN), a novel NMN based on compositions of Transformer modules.


**Summary Of The Review:**

The authors provide interesting observations, and the experiments are solid enough to show the effectiveness of the proposed method.

---

### Author Response · Authors · 2022-11-18
**To all: Thanks for the comments**

We thank the reviewers for their comments. We have run the suggested experiments and they all further strengthen the conclusions of our study. We have also rewrote a bit the paper to address all reviewer’s concerns. See updated version of the paper (all changes are indicated in blue, new experiments appear in the appendices). In the following, we summarize and explain all changes:

####  **1. Novelty (o4YA, pauq)**
>*“the paper directly substitutes the module in the NMN with the TRM encoder”*
*“The proposed model is a modular network composed of more expressive transformer modules.”*

This is a misunderstanding and we have clarified this in the revised paper (added few sentences at beginning of Section 3). Note that Transformers operate with tokens, and hence, modules in NMN can not be directly replaced by Transformer encoders. Transformer Module Network require to fully rearchitect NMNs in a new way with the philosophy of tokens in mind.

>*“Tree modular modular is not novel in NMN domain”*

The Tree NMN has been widely used in previous works. We have clarified this at the bottom of page 4.

#### **2. Contribution (zXs6)**
>*“The proposed method does not offer any fundamental contributions toward solving systematic generalization.”*

We politely disagree—we have demonstrated for the first time that the systematic generalization of Transformers in VQA can be substantially improved with a modular structure.

#### **3. Question parser (pauq)**
>*“Since all the experiments have a groundtruth program at test time, whether the question parser has the same systematic generalization ability is not verified.”*

We conducted experiments with ground-truth programs as we focus on analyzing systematic generalization capability of the models in terms of visual reasoning aspects. The development of a  good question parser (program generator) is a key open challenge as we stated in Limitations.

#### **4. What to compare our Transformer Module Networks with (o4YA, zXs6)**
- **Other types of CNN-based modules (o4YA)**
Bahdanau et al. 2020 showed that Vector-NMN is a CNN-based NMN that outperforms all previous modular approaches in terms of systematic generalization. Therefore we only conducted the comparisons with Vector-NMN on the synthetic datasets.

- **CFR [Nguyen et al. 2022] and HAN [Kim et al. 2020] (o4YA)**
Codes are not available and hence we can not obtain the systematic generalization accuracy of these methods (for CFR the code available is not to reproduce the results in the paper that we need, but for a challenge they tackled).

- **Standard Transformers with fewer layers (zXs6)**
We conducted new experiments with standard Transformers with fewer layers to confirm that the performance reported in the paper are not the result of overfitting. We added Table 12 in Appendix J in our revised paper. Table 12 shows that we cannot improve the systematic generalization performance by simply reducing the number of layers.

#### **5. Difference between the results with CLEVR-CoGenT and CLOSURE (pauq)**
>*“TMN can ignore the questions by accessing the groundtruth programs.”*

Note that we have thoroughly compared with baselines that also use groundtruth programs and TMN outperforms them.

>*“authors may verify this by training a new extractor from scratch similar to FiLM[1]”*

This was already done in D’Amario, et. al., “How modular should neural module networks be for systematic generalization?”, NeurIPS 2021, see Table App. 8 in Supplement of D’Amario et al. paper. This paper concludes that the configuration of the image encoder has a high impact in CLEVR-CoGenT. We clarified this in the paper at bottom of page 7.

#### **6. Further qualitative analysis (W6Kf)**
We added new Figs. 7, 8, and 9 in Appendix H and I in our revised paper to analyze how TMNs and Transformer w/ PR work. Figure 7 shows that modules in TMN work properly even when the input sequence of the sub-tasks are novel and unseen in the training. Thanks to the modularity, TMN leads to the correct answer on CLOSURE. Figure 8 and 9 show that Transformer w/ PR property operates the in-distribution program by assigning the sub-tasks into 12-layers while it fails to operate the novel program in CLOSURE by simply assigning the novel sequence of the sub-tasks in the same manner as the in-distribution program.

---

### Decision · Program_Chairs · 2023-01-20

**Decision:**

Reject

**Justification For Why Not Higher Score:**

Reviewers have critical concerns regarding the method and the experiments of this work, as mentioned above, and the author's responses do not resolve them.

**Justification For Why Not Lower Score:**

N/A

**Metareview: Summary, Strengths And Weaknesses:**

This paper aims to address the systematic generalization problem in VQA by combining the strengths of neural module networks and Transformers, and introducing a new Transformer Module Network (TMN). The TMN method is evaluated on three different VQA datasets.

Strengths:
- The systematic generalization problem is interesting and worth it to solve.
- The proposed method is technically sound and easy to understand.

Weaknesses:
- Reviewers have common concerns regarding the novelty of this work compared with the other NMN methods.
- It is unclear how the proposed method specifically addresses the systematic generalization problem.
- Lack of result comparison and ablation study.